# The Supportive Role of Plant-Based Substances in AMD Treatment and Their Potential

**DOI:** 10.3390/ijms26167906

**Published:** 2025-08-16

**Authors:** Karolina Klusek, Magdalena Kijowska, Maria Kiełbus, Julia Sławińska, Dominika Kuźmiuk, Tomasz Chorągiewicz, Robert Rejdak, Joanna Dolar-Szczasny

**Affiliations:** 1Student Scientific Club at the Department and Clinic of General and Paediatric Ophthalmology, Medical University of Lublin, Chmielna 1 Str., 20-079 Lublin, Poland; 61594@umlub.edu.pl (K.K.); 62102@umlub.edu.pl (M.K.); 62111@umlub.edu.pl (M.K.); 59775@umlub.edu.pl (J.S.); 58231@umlub.edu.pl (D.K.); 2Department of General and Paediatric Ophthalmology, Medical University of Lublin, Chmielna 1 Str., 20-079 Lublin, Poland; tomasz.choragiewicz@umlub.pl (T.C.); robert.rejdak@umlub.pl (R.R.)

**Keywords:** age-related macular degeneration, plant-derived substances used in the treatment of AMD, genetic background of AMD

## Abstract

There is growing interest in the use of natural plant-derived compounds, such as polyphenols (including curcumin), flavonoids, silymarin, anthocyanins, lutein, and zeaxanthin, for the treatment of age-related macular degeneration (AMD). These substances exhibit antioxidant, anti-inflammatory, and protective effects on retinal cells, contributing to the preservation of retinal integrity by modulating the key pathogenic mechanisms of AMD, including oxidative stress, chronic inflammation, and pathological neovascularization. Consequently, they hold potential to support conventional therapeutic approaches and slow disease progression. Current studies highlight their promising role as adjunctive agents in AMD management. This literature review provides a comprehensive analysis of the potential role of the aforementioned natural plant-derived compounds in the prevention and supportive treatment of age-related macular degeneration. It also discusses their natural sources, modes of administration and supplementation, and highlights the importance of a nutrient-rich diet as a key factor in maintaining ocular health. Furthermore, the review synthesizes current scientific knowledge on the ability of natural antioxidants to slow the progression of AMD and outlines future research directions aimed at improving diagnostic methods and developing more effective preventive and therapeutic strategies.

## 1. Introduction

Age-related macular degeneration (AMD) is a chronic, progressive retinal disorder affecting the central portion of the retina, i.e., the macula, which is responsible for high-resolution central vision [1,2,3,4]. AMD represents one of the leading causes of visual impairment among the elderly, particularly in highly developed countries [3,4,5,6,7,8,9].

The risk of developing the disease increases significantly after the age of 60 and continues to escalate with advancing age [6,10]. AMD not only leads to visual deterioration but also markedly affects patients’ daily functioning, limiting independence and increasing reliance on external assistance [5,7].

The diagnosis of AMD requires a comprehensive ophthalmic evaluation, including visual function assessments and advanced retinal imaging techniques. The core diagnostic methods include the following: dilated fundus examination to detect the presence of drusen and macular abnormalities, visual acuity testing, and optical coherence tomography (OCT), which enables a detailed cross-sectional analysis of the retinal layers. In suspected cases of exudative AMD, fluorescein angiography is particularly useful, allowing visualization of choroidal neovascularization (CNV) and vascular leakage [5].

Clinically, AMD is classified into two major forms: dry (non-exudative) and wet (exudative) [1,3,8]. The more prevalent, dry form progresses slowly and is characterized by gradual degeneration of the retinal pigment epithelium (RPE) and the accumulation of extracellular deposits, known as drusen, beneath the retina. The dry form may progress to the less common but more aggressive wet form, which is characterized by rapid vision loss and a high risk of irreversible blindness [1,4]. In wet AMD, abnormal choroidal neovascularization occurs, with leaking and bleeding vessels damaging the macula. This process is driven by the overexpression of VEGF, the key mediator of angiogenesis and vascular permeability [1,4,11].

The management of AMD depends on the clinical form and stage of the disease. In the case of dry AMD, treatment focuses on prevention and slowing disease progression. Dietary supplementation with antioxidants, vitamins (A, C, E), zinc, lutein, and zeaxanthin is recommended, as these agents may reduce oxidative stress and support retinal cell function [1,12,13,14,15]. However, no currently available therapy effectively reverses the retinal atrophy associated with advanced dry AMD [14,15,16,17,18]. Treatment of the wet form requires the intensive use of anti-VEGF drugs, which are administered intravitreally. These drugs inhibit abnormal neovascularization, and reduce retinal swelling and fluid leakage, thereby stabilizing or improving visual acuity. The most commonly used drugs are ranibizumab, aflibercept, and bevacizumab [1,16,17,19]. In addition to pharmacological approaches, ongoing research is exploring novel therapeutic strategies such as gene therapy, stem cell-based treatments, and advanced drug delivery systems, all aimed at enhancing treatment efficacy and safety [20].

## 2. Clinical Manifestations and Disease Course

According to epidemiological data, AMD occurs more frequently in women, particularly those with lighter iris pigmentation [6,7,10,21,22]. In the early stages, AMD is typically asymptomatic, especially if the pathological process affects only one eye. Subtle visual disturbances may remain unnoticed for a long time, leading to diagnosis often occurring in the intermediate or advanced stages [8,21,22]. As the disease progresses, characteristic clinical symptoms begin to emerge. One of the earliest noticeable signs is metamorphopsia, a distortion of vision in which straight lines appear wavy. Another warning sign is central scotoma, a dark spot in the central visual field, which gradually enlarges. With continued disease progression, visual acuity declines, contrast sensitivity decreases, and color perception is impaired. In advanced stages, the patient may lose the ability to perceive central details and see only a dark area, indicating the loss of functional central vision and severe impairment in daily activities [5,21,23].

## 3. Etiopathogenesis of Age-Related Macular Degeneration (AMD)

AMD is a complex retinal disease with a pathogenesis that is not fully understood. It involves an interplay of multiple factors, including oxidative stress, retinal pigment epithelium dysfunction, lipofuscin accumulation, inflammatory responses, and both genetic and environmental influences [4,8,11,24]. Figure 1 presents a schematic overview of the key pathogenic mechanisms involved in the development of age-related macular degeneration, as described in this section.

### 3.1. Oxidative Stress

The retina is one of the most metabolically active tissues in the human body due to its high photoreceptive activity and dense vascular network. This results in high oxygen consumption and an increased production of reactive oxygen species (ROS). When ROS production exceeds the antioxidant defense capacity, oxidative stress ensues, damaging RPE cells [7,8,9].

RPE cells play a pivotal role in retinal homeostasis, including the phagocytosis of photoreceptor outer segments. These segments are rich in long-chain polyunsaturated fatty acids (PUFAs), such as docosahexaenoic acid (DHA), which are particularly susceptible to peroxidation. Their oxidized products may accumulate as lipofuscin deposits [7,11,25].

Lipofuscin is a lysosomal, autofluorescent material that accumulates in long-lived cells like the RPE. While not inherently toxic, some of its components, especially bisretinoids like A2E, are phototoxic. Under UVA or blue light exposure, they generate ROS, thereby exacerbating oxidative stress in the retina and causing damage to lipids, proteins, and DNA [9,26].

Two primary mechanisms of ROS generation by lipofuscin are described in the literature: (1) photoreactivity of bisretinoids upon light exposure, leading to free radical formation; (2) biochemical synthesis of toxic bisretinoids such as A2E from N-retinylidene-N-retinylethanolamine, a product of the non-enzymatic condensation of retinal with phosphatidylethanolamine (PE) [9,11].

With age, antioxidant defense mechanisms become less effective, promoting lipofuscin accumulation and increasing the risk of AMD development [7,8].

Key antioxidant enzymes involved in protecting retinal cells from ROS include superoxide dismutase (SOD), catalase (CAT), and glutathione peroxidase (GPx). The reduced activity of these enzymes leads to oxidative damage accumulation [27]. Furthermore, antioxidant-rich diets, those containing lutein, zeaxanthin, and vitamins C and E, may alleviate oxidative stress and slow retinal degeneration [1,7,13,28].

### 3.2. Inflammatory Processes and Complement System Activation

Inflammatory mechanisms play a crucial role in AMD progression by disrupting immune homeostasis within the retina. The complement system, a component of innate immunity, normally functions to eliminate pathogens and damaged cells [8,24]. However, in AMD, chronic activation of the complement cascade leads to structural damage in the retina [8,24].

Dysregulation of the alternative complement pathway, often due to genetic polymorphisms, such as the Y402H variant in *the complement factor H (CFH)* gene, is particularly important. Reduced *CFH* activity impairs complement control, amplifying inflammatory responses in Bruch’s membrane and RPE cells [8,24,29].

Retinal inflammation is further exacerbated by the recruitment and activation of immune cells, including macrophages and microglia, which release pro-inflammatory cytokines that promote oxidative damage and cell death in photoreceptors and RPE [8,24,29].

### 3.3. Neovascularization

In the exudative (wet) form of AMD, choroidal neovascularization (CNV) is the principal pathological mechanism leading to vision loss. It involves the formation of abnormal blood vessels that penetrate Bruch’s membrane, causing fluid leakage, hemorrhages, and scarring.

The key driver of this process is vascular endothelial growth factor A (VEGF-A), whose expression is upregulated in response to hypoxia, oxidative stress, and inflammation. VEGF-A promotes endothelial proliferation and vascular permeability, facilitating CNV [8,24,29].

Angiopoietin-2 (Ang-2) acts synergistically with VEGF-A by destabilizing blood vessels and enhancing pathological angiogenesis [11,24,30].

## 4. Plant Substances with Potential in AMD Treatment

In recent years, more and more attention has been paid to natural substances derived from plants as a possible support for the treatment of age-related macular degeneration (AMD). Compounds such as polyphenols, flavonoids, carotenoids, or alkaloids exhibit a range of beneficial properties: they have antioxidant and anti-inflammatory effects and can also protect nerve cells. In this way, they can influence key processes responsible for the development of the disease, such as oxidative stress, damage to retinal pigment cells, or chronic inflammation. Table 1 presents key information on selected plant compounds with potential therapeutic use in the treatment of AMD.

### 4.1. Plant Compounds with Newly Revealed Therapeutic Potential in the Treatment of AMD

#### 4.1.1. Silymarin

Silymarin is a natural extract derived from the seeds of milk thistle (*Silybum marianum*), widely recognized for its potent antioxidant and anti-inflammatory activities, as well as its hepatoprotective effects [31]. Silibinin, as the main active component of silymarin, actively contributes to the therapeutic management of inflammatory disorders and oxidative stress-related damage, including age-related macular degeneration. AMD is associated with the degeneration of retinal cells, particularly retinal pigment epithelium, caused by excessive angiogenesis and oxidative stress. A key role in this process is played by VEGF, which stimulates the formation of new blood vessels. Silibinin acts on this mechanism by inhibiting VEGF secretion, which reduces abnormal angiogenesis and protects the retina from oxidative stress-induced damage [2,4].

Studies show that silibinin inhibits the secretion of vascular endothelial growth factor, one of the main factors responsible for neovascularization in the eye, leading to abnormal angiogenesis development [4]. Moreover, silibinin suppresses the PI3K/Akt/mTOR/p70S6K pathway, which is crucial in regulating angiogenesis and the response to oxidative stress [4,31]. By lowering the levels of HIF-1α, a protein involved in the hypoxia response, silibinin effectively reduces VEGF production in retinal pigment epithelial cells, thereby inhibiting new vessel formation in the eye [2,4].

By inhibiting the PI3K/Akt/mTOR pathway, silibinin regulates angiogenesis and the translation of HIF-1α, a key factor in VEGF expression and pathological neovascularization in AMD [4,31,32]. As a result, it decreases VEGF production and limits retinal edema and abnormal angiogenesis, which are the main causes of vision loss in wet AMD [32].

Animal model studies confirm that oral administration of silibinin effectively reduces neovascularization and retinal edema, as well as decreasing HIF-1α and VEGF levels under hypoxic conditions [2,32]. Additionally, silibinin increases the level of prolyl hydroxylase PHD2 and reduces the interaction of HIF-1α with the VHL protein, regulating HIF-1α degradation and stabilizing the homeostasis of this factor [32]. Furthermore, silibinin exhibits anti-inflammatory and anti-edematous effects in human retinal endothelial cells, counteracting excessive vascular permeability induced by diabetes and VEGF, which is significant for preventing diabetic retinopathy and diabetic macular edema [32].

In studies conducted on Brown Norway rats, AMD symptoms were induced by the intravitreal injection of VEGF and hypoxia exposure [4,31]. From day seven after injection, animals received oral silibinin at a dose of 500 mg/kg/day for three weeks [4]. This treatment effectively limited retinal edema, neovascularization, and vascular permeability, confirmed by fluorescein angiography [4]. Silibinin, a component of milk thistle, demonstrated the ability to stabilize HIF-1α protein, improving the ability of retinal pigment epithelial cells to adapt under hypoxia while simultaneously preventing excessive VEGF production, a factor playing a key role in AMD pathogenesis [4,31].

In the context of preventing disease progression, it has also been shown that silibinin reduces VEGF-induced retinal vascular permeability, as confirmed in studies on human retinal endothelial cells (HREC), where a concentration of 4 µg/mL significantly reduced this effect both under physiological and diabetic conditions [31]. This mechanism suggests that silibinin may stabilize the vascular endothelium, limiting its permeability and preventing pathological angiogenesis, making it a promising therapeutic option for treating neovascular AMD [31].

Oxidative stress plays a key role in AMD pathogenesis, leading to damage of retinal cells, including retinal pigment epithelial cells [2]. Silibinin, by neutralizing free radicals and reducing reactive oxygen species levels, may reduce retinal cell damage caused by oxidative stress [31]. Its mechanism involves an interaction with ROS by donating electrons or hydrogen atoms, leading to the stabilization of reactive molecules and preventing further damaging effects on cells. Thanks to its strong antioxidant properties, it effectively protects retinal cells from damage and supports the natural defense mechanisms of the visual system. This action is particularly important in neurodegenerative and ophthalmic diseases such as AMD [31]. It is worth emphasizing that it positively influences biochemical processes responsible for protection against oxidative stress, which may contribute to slowing disease progression and improving retinal function.

#### 4.1.2. Anthocyanins

Anthocyanins, natural flavonoids found, among others, in bilberry (*Vaccinium myrtillus*), exhibit a wide range of protective properties toward retinal cells [33,34]. Their action indicates significant potential as agents supporting both the prevention and treatment of age-related macular degeneration [35,36].

The mechanism of action of anthocyanins is primarily based on their strong antioxidant properties [33,35,36]. They neutralize free radicals and reactive oxygen species, which, in excess, damage lipids, proteins, and the DNA of retinal cells, leading to dysfunction and death of these cells [35]. By reducing oxidative stress, anthocyanins limit the cascade of cellular damage, which is crucial for protecting the delicate retinal structure [35]. Anthocyanins also inhibit the inflammatory response in retinal tissue, which is important in the context of chronic inflammation, a key factor in AMD progression leading to the degradation of retinal cells and their microenvironment. By modulating inflammatory pathways, they reduce the production of pro-inflammatory cytokines, thus supporting the maintenance of homeostasis and tissue health [35].

An in vitro study conducted by Bornsek at al. has shown that anthocyanins found in bilberries and blueberries have strong antioxidant activity at the cellular level, even at very low concentrations (<1 μg/L) [36]. This was demonstrated using the Cellular Antioxidant Activity (CAA) assay on various cell lines such as colon cancer cells, liver cells, endothelial cells, and the vascular smooth muscle cells of rats [36]. Despite the low bioavailability of anthocyanins, their strong antioxidant activity in cells forms the basis of their beneficial health effects, including their potential use in AMD treatment.

An experimental study by Wang et al. evaluated the effect of a bilberry anthocyanin extract (BAE) in a model of light-induced retinal degeneration in pigmented rabbits [35]. The animals were exposed to light at an intensity of 18,000 lx for 2 h, then orally administered BAE for seven days at doses of 250 or 500 mg/kg [35]. Retinal function was assessed using electroretinography (ERG), along with histological and biochemical changes [35]. The results showed that BAE improves photoreceptor function, protects retinal structure, increases antioxidant enzyme activity, and reduces oxidative stress markers. Moreover, anti-inflammatory effects were observed through reduced levels of IL-1β and VEGF and the modulation of apoptosis, including the increased expression of Bcl-2 and decreased levels of Bax and caspase-3 [35]. These data suggest that anthocyanins from blueberries may act as protective agents against retinal damage induced by light exposure.

Another important aspect of their action is their ability to inhibit apoptosis, i.e., programmed cell death [2,35]. Studies have shown that anthocyanins limit the activation of apoptotic cascades in retinal cells, allowing the maintenance of their number and function, counteracting cell loss characteristic of this condition [35].

As a result of the mechanisms described above, anthocyanin supplementation protects the retina from damage caused by light and oxidative stress, prevents excessive inflammatory processes, and inhibits cell death [35]. This makes it possible to slow the progression of macular degeneration, preserve visual function, and improve the quality of life for patients at risk of AMD [35].

Similar properties are shown by anthocyanins contained in extracts from the American blueberry (*Vaccinium angustifolium*), which also possesses strong antioxidant and anti-inflammatory activities [34,37]. In vitro studies and animal models have demonstrated that these anthocyanins effectively neutralize free radicals and reduce reactive oxygen species production in retinal cells [37]. Their anti-inflammatory effect is manifested by the inhibition of pro-inflammatory cytokine production and modulation of signaling pathways responsible for inflammation, which is crucial in limiting AMD progression [34,37]. Moreover, anthocyanins from the American blueberry protect retinal cells from apoptosis, supporting the survival of photoreceptors and retinal pigment epithelium [37]. The multifunctional antioxidant, anti-inflammatory, and anti-apoptotic effects make these compounds a promising agent in AMD prevention and therapy [33,34,37].

### 4.2. Lutein and Zeaxanthin

About 20 different types of carotenoids have been identified in human plasma and serum, most of which are nonpolar carotenoids called carotenes, such as β-carotene, α-carotene, and lycopene. In contrast, lutein and zeaxanthin, belonging to the xanthophyll group, are present in much smaller amounts in the bloodstream but show a distinct selective accumulation in eye tissues, particularly in the retina and lens [1,13,38,39,40].

In the retina of the human eye, especially in the macula lutea, lutein and zeaxanthin form the so-called macular pigment, which plays a key role in protecting against the harmful effects of blue light and in neutralizing free radicals [39,40]. The concentration of these xanthophylls in the retina is many times higher than in other body tissues [38,41]. In the central part of the macula (the foveal center), zeaxanthin predominates, whereas lutein dominates in the peripheral regions of the retina [12,28,42,43,44].

It is important to note that lutein and zeaxanthin are not endogenously synthesized by the human body and must be obtained from the diet. The main sources of these carotenoids are green leafy vegetables such as spinach and kale, as well as corn and egg yolks [1,45,46]. However, despite their dietary presence, plasma concentrations of lutein and zeaxanthin are relatively low, emphasizing the importance of their selective transport and accumulation in eye tissues [28,38,40,45,46].

In a study published by Wu et al. (2015) [47], a prospective cohort analysis was conducted involving over 100,000 participants (during the study, 1361 were diagnosed with intermediate AMD and 1118 with advanced, mostly neovascular AMD cases) over a 20-year follow-up period. It was demonstrated that a higher intake of lutein and zeaxanthin was associated with significant risk reduction for developing advanced AMD.

Similar results were obtained in the study by Cho et al. (2008) [48], which also found a significant reduction in the risk of neovascular AMD among individuals in the highest quartile of lutein and zeaxanthin intake compared to those with the lowest intake. However, the data did not confirm the protective role of lutein and zeaxanthin intake regarding the risk of early AMD.

Nevertheless, a randomized, double-blind, placebo-controlled trial conducted by Huang et al. in 2014 [49] provided evidence for the beneficial effects of lutein and zeaxanthin supplementation on retinal function in patients with early AMD. After 12 months of supplementation, improvements were observed in electroretinographic parameters and increased macular pigment optical density (MPOD). These results suggest that lutein and zeaxanthin may not only act protectively but also support functional retinal regeneration.

Meanwhile, a study conducted by Sawa et al. (2020) [50], known as the Sakai Lutein Study, analyzed the effect of lutein supplementation in Japanese patients with unilateral AMD. The study involved 39 patients who were randomly assigned to one of two groups receiving lutein for six months, in the form of capsules containing either beeswax or glycerol fatty acid esters. After six months of supplementation with 20 mg of lutein daily, an increase in plasma lutein concentration was observed, and, in the group receiving the beeswax-based preparation, a significant increase in MPOD was noted after three months of supplementation, although the increase from month three to six was only slightly greater.

In summary, although many epidemiological and interventional studies indicate the potentially beneficial effect of lutein and zeaxanthin on macular health and their role in preventing and supporting the treatment of age-related macular degeneration, the results are not entirely consistent [28,38,47,48,49,50]. Their mechanisms of action include antioxidant properties, the ability to modulate inflammatory responses, and protection against phototoxic damage [45,46]. Nonetheless, some studies do not unequivocally confirm their effectiveness, particularly in the context of early AMD stages [48,50]. Despite these discrepancies, supplementation with these carotenoids provides grounds for hope in slowing disease progression. Further research is needed to conclusively confirm their efficacy and determine optimal therapeutic doses.

### 4.3. Polyphenols

Polyphenols neutralize reactive oxygen species, inhibit lipid peroxidation, reduce inflammation, and modulate the expression of genes involved in angiogenesis and the apoptosis of retinal cells [35]. Their action also includes effects on signaling pathways related to AMD pathogenesis, such as PI3K/Akt/mTOR and Nrf2/HO-1 [51].

Epigallocatechin-3-gallate (EGCG), the main catechin of green tea, exhibits strong antioxidant and anti-inflammatory effects. The mechanisms of EGCG action include neutralization of reactive oxygen species, which reduces oxidative stress in retinal pigment epithelium cells. Additionally, EGCG inhibits the TLR4 signaling pathway via a 67LR receptor-dependent mechanism, leading to decreased expression of pro-inflammatory factors such as NF-κB and AP-1. Moreover, EGCG increases cAMP levels in endothelial cells, activating protein kinase A (PKA) and resulting in the phosphorylation of eNOS and VASP, influencing vascular functions [52,53]. Studies suggest that EGCG protects RPE cells from apoptosis, reduces TNF-α and IL-6 expression, and decreases oxidative damage in AMD models. Furthermore, it supports blood–retina barrier integrity and improves photoreceptor survival [51,54]. In the context of AMD therapy, EGCG shows potential as an adjunctive treatment. In vitro studies have demonstrated that EGCG reduces H_2_O_2_-induced apoptosis in RPE cells by modulating the Cyfip2/AKT pathway, suggesting a protective role in AMD development [54]. Additionally, EGCG may modulate signaling pathways related to autophagy and apoptosis through interactions with mTOR and ULK1, which may affect the balance between these processes in retinal cells [51,55]. Although most EGCG research in AMD has been performed in vitro or in animal models, there is a need for controlled clinical trials to evaluate the efficacy of EGCG supplementation or green tea consumption as a complement to anti-VEGF therapy in the exudative form of AMD [51].

Resveratrol, found in grape skins and red wine, exhibits antioxidant, anti-inflammatory, and neuroprotective effects. Its mechanisms include activation of sirtuin 1 (SIRT1) deacetylase, leading to improved mitochondrial function and reduced oxidative stress, the inhibition of vascular endothelial growth factor expression, which may limit neovascularization in exudative AMD, and the modulation of apoptosis-related signaling pathways that protect RPE cells from damage [56,57]. Clinical studies showed that supplementation with Resvega (containing resveratrol, among other compounds) combined with aflibercept treatment (IAI—intravitreal anti-VEGF injections) yielded significantly better results than aflibercept monotherapy. Patients experienced improved visual acuity, increased contrast sensitivity, and a reduced number of required injections [58]. Additionally, studies suggest that daily oral intake of Resvega in patients with exudative AMD can be considered a useful adjunct to established therapy, enhancing combined treatment efficacy. This regimen improved injection frequency and positively affected fibrosis prevention [59]. Despite these promising clinical observations, several important limitations remain. The majority of available data on resveratrol stems from combination therapies such as Resvega, making it difficult to isolate its individual contribution to observed clinical outcomes. Moreover, variability in formulations, dosing, and study complicates cross-study comparisons. While resveratrol has shown biological activity on key AMD-related pathways, such as oxidative stress and VEGF expression, the long-term safety, optimal dosage, and effectiveness of resveratrol monotherapy remain unclear. Further randomized controlled trials are necessary to determine whether resveratrol alone can provide clinically significant benefits in AMD, particularly in comparison to existing anti-VEGF therapies [56,57,58,59].

Chlorogenic acid (CGA), an ester of caffeic and quinic acids, present in coffee and many fruits, exhibits strong antioxidant and anti-inflammatory properties. Its mechanisms include scavenging reactive oxygen species, such as superoxide anions and hydroxyl radicals, leading to reduced oxidative stress in retinal cells [60]. Furthermore, CGA inhibits the NF-κB signaling pathway, resulting in decreased expression of pro-inflammatory cytokines like interleukin-8 (IL-8), contributing to inflammation reduction in retinal tissues. Additionally, CGA can form complexes with proteins via hydrogen bonding, potentially protecting retinal cells from oxidative damage [60,61]. In vitro studies showed that CGA inhibits VEGF expression, a key mediator of neovascularization in AMD, and supports blood–retina barrier integrity, potentially counteracting pathological retinal neovascularization. Animal models demonstrated CGA’s ability to inhibit choroidal neovascularization, a critical factor in exudative AMD pathogenesis [62]. CGA administration reduced VEGF expression and limited new blood vessel growth in the retina. Despite promising preclinical results, clinical data confirming CGA’s efficacy in AMD therapy in humans are currently lacking [63]. Further research is necessary to define CGA’s potential role as a complement to existing AMD treatments. Several gaps remain in our understanding of CGA’s therapeutic applicability in AMD. Most evidence stems from in vitro and animal studies, often employing non-physiological concentrations or delivery methods that may not be feasible in human application. It also remains uncertain whether CGA’s antiangiogenic effects are robust enough to exert clinically relevant outcomes when used alone. The lack of human trials limits conclusions on safety, efficacy, and optimal dosing. Furthermore, interindividual variability in CGA metabolism and the impact of gut microbiota on its biotransformation introduce additional complexity that needs to be addressed in future investigations [62,63]. Most polyphenol studies focus on oral supplementation, but a limitation of this route is low bioavailability caused by rapid hepatic metabolism and poor water solubility [60]. To address these issues, carrier systems such as liposomes, nanoparticles, and cyclodextrins are being developed to enhance retinal penetration and prolong activity [64]. Topical administration of polyphenols is also being explored, e.g., EGCG as an ingredient in eye drops could provide high local active substance concentration while limiting adverse effects [60].

#### Curcumin

One of the pathomechanisms involved in pathology development within the retinal pigment epithelium in AMD is impaired autophagy. This process, responsible for the degradation of damaged organelles, unnecessary cytoplasmic components, and other protein aggregates, is crucial for maintaining physiological homeostasis within the cell [33,34]. In recent years, it has been shown that functional disturbances of autophagy constitute one of the key factors in the development of age-related macular degeneration [8,65,66,67,68]. They lead to the formation of extracellular deposits (so-called drusen), aggregation of, among others, lipofuscin, and damage to and atrophy of RPE cells [67,68,69].

Among natural plant compounds showing potential as safe and effective autophagy activators, curcumin draws particular attention. It is a substance obtained from the rhizome of turmeric (*Curcuma longa*), a plant belonging to the ginger family (*Zingiberaceae*) [70,71]. Curcumin, also known as diferuloylmethane (E100, European Food Safety Authority designation), is a chemical compound with the formula 1,7-bis-(4-hydroxy-3-methoxyphenyl)-1,6-heptadiene-3,5-dione [72]. The anti-inflammatory and antioxidant effects of curcumin have also been demonstrated in other retinal diseases such as diabetic retinopathy, retinitis pigmentosa, retinoblastoma, and proliferative vitreoretinopathy [70,71,72,73].

Pinelli et al. (2023) [74] showed that curcumin can completely prevent the loss of RPE cells and other pathological changes induced by 3-methyladenine (3-MA), which was used in the study as an autophagy inhibitor. This protein acts by blocking the activity of class III PI3K kinase, leading to the inhibition of autophagosome formation [75]. 3-MA has a confirmed role in the degenerative processes of RPE cells, loss of cell integrity, and apoptosis [69,75,76]. Experimental studies showed that curcumin effectively prevents the loss of retinal pigment epithelial cells and the development of pathological changes induced by 3-MA at concentrations of 10 mM and 20 mM. Its protective action was associated with the restoration of the correct number of LC3-positive autophagosomes and maintenance of the structural integrity of intercellular junctions, which was confirmed by immunohistochemical analyses and microscopic observations [74].

Beyond disturbances in autophagy processes, numerous scientific studies confirm a significantly increased activity of the Wnt signaling pathway in the pathogenesis of age-related macular degeneration, suggesting its key role in the molecular mechanisms underlying this disease development [77,78,79].

The Wnt pathway is a complex and multifunctional cellular signaling pathway involved in regulating cell proliferation, differentiation, and survival, as well as maintaining stem cell homeostasis [77,80].

For canonical Wnt pathway activation, Wnt ligands, glycosylated cysteine-rich proteins, must bind to frizzled (Fz) receptors and LRP5/6 co-receptors. This leads to inhibition of the complex responsible for β-catenin degradation, enabling its accumulation in the cytoplasm and translocation to the nucleus. There, β-catenin binds to transcription factors TCF/LEF and activates gene expression related to, among others, angiogenesis and inflammation, such as *VEGF*, *c-Myc*, *cyclin D1*, *ICAM-1*, *TNF-α*, *CTGF*, and *HIF-1* [77,78,81].

Aberrant activation of the Wnt pathway can therefore contribute to the development of pathological angiogenesis and inflammation observed in the exudative form of AMD. For this reason, this pathway represents a potential therapeutic target, which has already been confirmed by reports showing the effectiveness of antibodies blocking LRP6 in rat and mouse CNV models [78].

Although many studies confirm the direct inhibitory effect of curcumin on the Wnt/β-catenin pathway in gastrointestinal cancer therapy, scientific reports concerning the direct use of curcumin in the modulation of this pathway in eye diseases such as macular degeneration are still relatively few [82,83,84]. However, studies confirm that curcumin affects pro-inflammatory factors, increasing oxidative stress and apoptotic processes, which are closely related to Wnt pathway activation [78,85,86,87].

Chen et al. (2006) [85], in studies on U937 and Raji cell lines, described that TNF-α increases VEGF expression in cancer cells, and curcumin inhibits this expression and angiogenesis in human umbilical vein endothelial cells (HUVECs). *VEGF* is a direct target gene of the Wnt/β-catenin pathway, and TNF-α, also activated by this pathway, can further enhance angiogenesis through inflammatory pathways [78]. By lowering levels of both VEGF and TNF-α, curcumin exerts antiangiogenic and anti-inflammatory effects, which may result, among others, from its influence on inhibiting β-catenin activity in the nucleus and blocking the expression of these target genes [8,65,85,86].

Moreover, it has been shown that besides antiangiogenic effects, curcumin, its metabolites tetrahydrocurcumin and octahydrocurcumin, as well as the prodrug curcumin diethyl disuccinate (CurDD), protect retinal pigment epithelial cells against oxidative stress by increasing expression of antioxidant enzymes HO-1 and NQO1 and modulating apoptotic proteins Bax and Bcl-2 [86,87]. Through its antioxidant activity, curcumin may indirectly inhibit excessive Wnt pathway activation by limiting ROS, which themselves can initiate this signaling pathway [81,88].

Although numerous in vitro studies support the role of curcumin in modulating autophagy and inflammatory signaling in retinal pigment epithelial (RPE) cells, the translational validity of these findings remains limited. Most studies utilize supraphysiological concentrations (e.g., 10–20 mM) that are unlikely to be achieved via standard dietary intake or topical administration. Furthermore, the precise molecular mechanism by which curcumin may inhibit pathological angiogenesis in AMD, whether directly through the Wnt/β-catenin pathway or indirectly via oxidative stress reduction, remains unresolved. No clinical trials have yet confirmed curcumin’s efficacy in AMD, highlighting a major gap between preclinical efficacy and real-world therapeutic applicability [74,78,85,86,87].

In summary, polyphenols including EGCG, resveratrol, curcumin and chlorogenic acid exhibit multifunctional protective effects on the retina, neutralizing oxidative stress, inhibiting inflammation, and limiting angiogenesis via influence on PI3K/Akt/mTOR and Nrf2/HO-1 pathways [7,51]. Although in vitro and in vivo results are promising, especially regarding RPE cell protection and VEGF inhibition, clinical confirmation of their efficacy is still lacking [51,60]. The development of advanced delivery systems, including EGCG-containing eye drops, may enhance their therapeutic potential in AMD treatment [60,64].

### 4.4. Flavonoids

Flavonoids, plant secondary metabolites, possess high therapeutic potential mainly due to their antioxidant, anti-inflammatory, and neuroprotective properties [89,90]. Among flavonoids, kaempferol, baicalin, and genistein stand out for their multifunctional protective actions in the retina.

Kaempferol is a naturally occurring flavonoid found in cruciferous vegetables, berries, and tea. In the context of age-related macular degeneration, it exhibits strong antioxidant properties and protects retinal pigment epithelium cells [91,92]. Its mechanism mainly involves inhibiting apoptosis induced by oxidative stress through modulation of apoptosis-related gene expression, including the regulation of the Bax/Bcl-2 protein ratio and reduction of caspase-3 activity [92,93]. Simultaneously, kaempferol decreases VEGF levels and blocks the PI3K/Akt signaling pathway, which plays a key role in the pathological angiogenesis seen in neovascular AMD. Experimental data confirm that kaempferol effectively protects RPE cells from oxidative damage, supporting their survival by activating molecular defense mechanisms [92,93]. Moreover, animal studies show kaempferol limits the photoreceptor degeneration caused by excessive light exposure, further supporting its therapeutic potential in AMD [94]. Clinically, kaempferol’s oral administration is limited by low bioavailability. Therefore, new delivery technologies are actively being developed to enhance its therapeutic efficacy, including nanoemulsions, liposomes, and other nanocarrier systems [95,96,97]. It has been demonstrated that kaempferol nanoemulsions significantly improve its solubility in aqueous environments and chemical stability, favoring better bioavailability and enabling controlled, targeted release to retinal tissues [95]. Another approach involves nanostructured lipid carriers modified with hyaluronic acid (HA-KA-NLCs), which show an increased capacity for efficient kaempferol transport to retinal cells, this being potentially important therapeutically for AMD [96]. Additionally, lipid nanoparticles capable of crossing the blood–retina barrier to deliver kaempferol directly to the retina show promise for increasing local efficacy while reducing systemic side effects [97].

Baicalin, a flavonoid isolated from the root of *Scutellaria baicalensis*, exhibits broad anti-inflammatory and antioxidant effects, giving it therapeutic potential in age-related macular degeneration [98,99,100,101]. In vitro studies on retinal pigment epithelium cells (ARPE-19) showed that baicalin reduced β-amyloid (Aβ) toxicity, a factor linked to dry AMD pathogenesis [102]. The mechanism involved suppressing NLRP3 inflammasome activity by increasing miR-223 expression, thereby limiting inflammation and pyroptosis [98]. Furthermore, in rat models of glutamate-induced retinal ganglion cell (RGC) damage, 10 µM baicalin significantly improved R28 cell survival by reducing reactive oxygen species, increasing antioxidant enzyme activity and suppressing pro-inflammatory factors such as iNOS, TNF-α, IL-6, and IL-1β. In vivo, baicalin increased retinal ganglion cell layer thickness and reduced overall retinal thinning in a dose- and time-dependent manner. A key mechanism was the inhibition of the JAK/STAT signaling pathway, which is important in inflammation and apoptosis [100]. Clinically, oral or systemic administration of flavonoids like baicalin faces issues of low bioavailability and limited retinal transport. Recently, a novel in situ gel formulation containing a baicalin nanoemulsion for ocular surface application was developed. This form increased corneal permeability and prolonged baicalin release within ocular structures, significantly reducing inflammatory markers (TNF-α, IL-6), oxidative stress, and apoptosis in a dry AMD mouse model [103]. Literature reviews indicate flavonoids from *Scutellaria baicalensis*, including baicalin, also have neuroprotective properties, making them attractive for multi-targeted retinal disease therapy. Their activity involves the regulation of NF-κB and Nrf2 pathways and inhibition of microglial activation, which may support retinal structural integrity in AMD patients [99,100].

Genistein is an isoflavone naturally found in soy (*Glycine max*), known primarily for its anti-inflammatory, antioxidant, and antiangiogenic properties [104,105]. A key mechanism is its ability to inhibit angiogenesis. In an animal AMD model, genistein significantly reduced the size of choroidal neovascularization (CNV) lesions by lowering levels of the MCP-1, ICAM-1, and MMP-9 proteins involved in inflammation and angiogenesis [106,107]. It also exerts strong anti-inflammatory effects by inhibiting signaling pathways such as NF-κB and by reducing pro-inflammatory cytokine production and reactive oxygen species. These properties may protect retinal pigment epithelium cells from inflammation-related damage in AMD [108,109]. Genistein is known as a selective protein tyrosine kinase (PTK) inhibitor. It competitively binds to the ATP-binding site in the catalytic domain of tyrosine kinases, inhibiting tyrosine phosphorylation in signaling proteins. This interrupts signaling through pathways, such as EGFR, PDGFR, Src, Fyn, and PI3K/AKT, critical for cell proliferation, migration, and angiogenesis [110]. By inhibiting these kinases, genistein may limit the uncontrolled angiogenesis characteristic of neovascular AMD [106]. Currently, genistein is under investigation regarding various delivery methods that may enhance its therapeutic efficacy in AMD treatment. Oral supplementation remains the most common due to ease of administration and availability. Pharmacokinetic studies show genistein is rapidly absorbed, reaching peak plasma concentrations within hours; however, its bioavailability is limited by extensive first-pass metabolism and rapid elimination, which may reduce therapeutic potential [111]. To improve efficacy, intense research focuses on novel delivery systems such as nanotechnology-based carriers including lipid nanoparticles, liposomes, and nanoemulsions. These can significantly enhance genistein’s aqueous solubility and chemical stability, and allow controlled release of the active compound. Numerous nanoformulations and their potential applications in treating ocular diseases, including AMD, have been reported [112,113].

In summary, flavonoids such as kaempferol, baicalin, and genistein exhibit diverse mechanisms of action in AMD pathogenesis: antioxidative, anti-inflammatory, antiapoptotic, and antiangiogenic [4,8,11,24,104]. Their combined activities, modulating key signaling pathways such as PI3K/Akt/mTOR, NF-κB, Nrf2, and JAK/STAT, highlight their potential as adjunctive agents in AMD therapy. These complex interactions are visually summarized in Figure 2, which illustrates the modulatory effects of selected flavonoids on key biochemical pathways involved in AMD. However, a major challenge is their low bioavailability and limited retinal penetration, which are actively addressed by advanced drug delivery systems, including nanoformulations and ocular gels [95,96,97,103]. Clinical data on efficacy in AMD patients remain limited, emphasizing the need for well-designed trials [90].

**Table 1 ijms-26-07906-t001:** Selected Plant-Based Substances with Therapeutic Potential in AMD.

Substance Name	Source of Substances	Mode of Action	Clinical Evidence	Comments
Curcumin		Activation of autophagy [74]		
	Turmeric rhizome (*Curcuma longa*)	Antiangiogenic [85]	In vivo [86] In vitro [74,85]	Curcumin’s effects were observed even at a relatively low concentration of 10 μM, suggesting its high efficacy and therapeutic potential [74]
		Anti-inflammatory [85,86]		
Sylibinin	Milk thistle (*Sylibum marianum*)	Antioxidant [4]	In vivo & in vitro [4]	An interesting aspect of silybinin’s action is that it increases HIF-1α protein levels without affecting its mRNA, indicating regulation at the level of protein translation or stability, rather than transcription
		Antiangiogenic [4]		
Anthocyane	Blueberry (*Vaccinium angustifolium*)*,* Blueberry (*Vaccinium myrtillus*)	Anti-inflammatory [35,37]	In vivo [35]	Studies indicate that anthocyanins can penetrate the blood–retina barrier, allowing them to have a direct protective effect on retinal cells in AMD
		Antioxidant [35,37]		
Epigallocatechin-3-gallate (EGCG)	Green tea	Antioxidant [52,54]	In vitro [54]	It also affects the regulation of the signaling pathways responsible for cell apoptosis, which may contribute to protecting the retina from degeneration
Resveratrol	Grape skins, red wine	Antioxidant [56,57] Anti-inflammatory [56,57] Neuroprotective [56,57]	In vivo [56,57]	
Chlorogenic acid (CGA)	Green coffee, Jerusalem artichoke, blueberries	Antioxidant [60] Anti-inflammatory [61,62]	In vivo [61,62]	In animal models, CGA has shown the ability to inhibit choroidal neovascularization
Kaempferol	Brassica vegetables, berries, tea	Antioxidant [91,92] Anti-inflammatory [91,92] Neuroprotective [94]	In vitro [91,92] In vivo [94]	There is research into new delivery systems for kaempferol, such as nanoemulsions, which improve its bioavailability and may increase its therapeutic efficacy
Baicalin	Root of *Scutellaria baicalensis*	Antioxidant [98,99,100,101] Anti-inflammatory [98,99,100,101] Neuroprotective [99,100]	In vivo [98] In vivo & in vitro [100]	A recent study developed a new form of baicalin administration, an in situ gel containing a nanoemulsion of the active ingredient
Genistein	Common soybean (*Glycine max*)	Anti-inflammatory [104,105] Antioxidant [104,105] Anti-angiogenic [106,107]	In vivo [106,107]	It is known as a selective tyrosine kinase (PTK) inhibitor

## 5. Limitations

Plant-derived substances such as carotenoids, polyphenols, flavonoids, and anthocyanins exhibit anti-inflammatory, antioxidant, and neuroprotective properties in laboratory studies, which may counteract the pathogenesis of age-related macular degeneration associated with chronic oxidative stress and retinal pigment epithelium dysfunction. However, researchers continue to debate their actual effectiveness in humans, mainly due to the limited quality and inconsistency of available scientific data.

Many clinical studies remain observational and involve small patient cohorts, which increases the risk of statistical errors, limits analytical power, and reduces the reliability of therapeutic effect results.

Researchers primarily conduct studies on polyphenols and flavonoids using in vitro and animal models, with relatively few clinical trials. These clinical trials often present variable methodological quality and small sample sizes. Such methodological limitations significantly hinder the extrapolation of results to the general population and the reliable assessment of the true therapeutic efficacy of these substances in humans.

Furthermore, the chemical forms and concentrations of compounds in both plant sources and supplement formulations vary widely. The lack of standardization in dosing, pharmaceutical forms, extraction methods, and chemical analyses prevents direct comparisons between studies and complicates drawing definitive conclusions regarding their efficacy.

Low bioavailability also poses a major limitation for many plant-derived substances, such as curcumin and silymarin, substantially reducing their biological activity within the body and impacting their in vivo efficacy.

Moreover, pharmacokinetic and pharmacodynamic interactions between various plant components, as well as between these components and the medications used by patients with macular degeneration, may significantly alter therapeutic outcomes, leading to enhancement, reduction, or adverse effects.

Most studies also fail to include long-term patient follow-up, which is particularly important in chronic degenerative diseases like age-related macular degeneration.

A diet rich in antioxidants and polyphenols constitutes a complex dietary pattern whose impact on the development and progression of macular degeneration is modulated by numerous factors such as genetic predisposition, lifestyle, environment, and comorbid chronic diseases.

The complexity of interactions between dietary components and the individual biological profile of patients complicates the identification of the specific effects of individual bioactive compounds.

Additionally, individual factors including genetic polymorphisms, disease stage, comorbidities, and metabolic rate differences influence both the efficacy and safety of plant-derived substances. Therefore, researchers need further well-designed, randomized clinical trials involving large populations with long-term monitoring of therapeutic outcomes.

Although preclinical investigations have provided valuable mechanistic insights into the potential therapeutic effects of curcumin and related phytochemicals in AMD, a critical evaluation reveals several translational gaps. In vitro models such as ARPE-19 cells lack the structural and physiological complexity of the native retina, including immune cell interactions and extracellular matrix dynamics. Similarly, animal models differ from humans in retinal anatomy, drug metabolism, and inflammatory signaling, often resulting in overestimated efficacy. Many experimental protocols also involve non-clinical dosing strategies or delivery routes that are not applicable in humans. Moreover, poor systemic bioavailability remains a major barrier for compounds like curcumin, limiting their effectiveness under physiological conditions despite promising laboratory results. These constraints underscore the need for rigorous human studies with appropriate design, dosing, and long-term follow-up to validate the therapeutic relevance of plant-derived compounds in AMD.

## 6. Conclusions

A diet characterized by a high intake of plant-based foods, such as vegetables, fruits, legumes, whole grains, and nuts, along with regular consumption of fish and healthy fats like olive oil, demonstrates a significant protective effect on the structures of the retina. Numerous epidemiological and clinical studies indicate that adherence to a diet based on these principles is associated with a reduced risk of developing age-related macular degeneration and with slower disease progression.

The protective mechanisms are attributed to the presence of various bioactive compounds such as polyphenols, carotenoids (including lutein and zeaxanthin), omega-3 fatty acids, vitamins C and E, and trace elements such as zinc and selenium. These substances exhibit strong antioxidant and anti-inflammatory properties, allowing them to reduce oxidative stress, neutralize reactive oxygen species, and suppress chronic inflammatory processes within the retina, all of which plays a key role in AMD pathogenesis.

Components of a diet rich in antioxidants and beneficial fats support the integrity of the blood–retina barrier, improve the function of the retinal pigment epithelium, and may reduce the accumulation of lipofuscin and drusen that is characteristic of AMD. For these reasons, this diet represents an important element of AMD prevention and therapy support, and its implementation may serve as an effective and safe tool for delaying the progression of this degenerative disease.

Plant-derived substances such as polyphenols, flavonoids, and carotenoids show significant therapeutic potential in the treatment of age-related macular degeneration, mainly due to their antioxidant, anti-inflammatory, and neuroprotective properties. Compounds such as resveratrol, curcumin, quercetin, luteolin, and epigallocatechin gallate (EGCG) modulate key molecular pathways involved in AMD pathogenesis, including oxidative stress, apoptosis of retinal pigment epithelium cells, and angiogenesis driven by vascular endothelial growth factor.

Results from in vitro and in vivo studies confirm the ability of these compounds to protect retinal pigment epithelium cells from oxidative damage and to inhibit pathological neovascularization, which may delay the progression of both dry and wet forms of the disease. However, the therapeutic effectiveness of these substances may be limited by their low bioavailability and limited stability, posing a challenge for their practical application. Additionally, the limitations of research on plant-derived substances in the treatment of age-related macular degeneration primarily include the low quality and inconsistency of available scientific data, small patient cohorts in clinical studies, and the lack of standardization of preparations. The absence of long-term patient follow-up hinders a reliable assessment of their efficacy and safety, significantly complicating the formulation of definitive therapeutic conclusions.

A promising solution is the development of modern delivery systems, such as nanocarriers or liposomal formulations, which increase their bioavailability and efficacy (AMD). Based on this, the use of plant-derived substances may serve as a safe and supportive therapeutic approach in AMD treatment, particularly as part of combination therapy with conventional methods.

Recently, substances considered “novel” in the context of AMD treatment are also gaining attention. Particularly promising are silibinin and anthocyanins, whose supplementation has provided evidence for their potential use in AMD therapy through inhibition of pathological angiogenesis, the PI3K/Akt/mTOR pathway, and the reduced expression of VEGF and HIF-1α (silibinin), as well as antioxidant and anti-inflammatory effects, resulting in increased retinal cell integrity and reduced hypoxia- and oxidative-stress-induced damage (anthocyanins).

## 7. Materials and Methods

A comprehensive literature search in English was conducted using the PubMed, ResearchGate, and Google Scholar databases, covering publications from 21 April 2009 to 4 December 2024. The following keywords were used in the search: “Anthocyanins in the treatment of AMD,” “AMD pathogenesis and symptoms,” “plant-derived substances used in the treatment of AMD,” “genetic background of AMD,” and “Mediterranean diet in the treatment of AMD”.

The initial search yielded a large number of results, which were then screened for relevance based on title and abstract analysis. Studies focusing on the effects of silybin and curcumin on the pathogenetic mechanisms of age-related macular degeneration, particularly their protective effects on retinal cells and the modulation of processes such as angiogenesis, oxidative stress, and autophagy, were included in the analysis. Publications evaluating the impact of polyphenols and flavonoids, especially regarding their neuroprotective potential for retinal cells in the context of molecular mechanisms, were also considered. These studies encompassed both experimental models and in vitro analyses, allowing the assessment of the efficacy of these substances in preventing the degeneration of retinal pigment epithelium cells. Special attention was given to the molecular mechanisms through which silybin and curcumin may slow disease progression and protect against vision loss. Exclusion criteria included publications in languages other than English and studies not directly related to AMD or the molecular analysis of retinal changes.

The search process and selection strategy are summarized in Table 2 and illustrated in Figure 3. In total, 114 articles were selected after title and abstract screening, duplicate removal, and full-text assessment for eligibility. Bibliographic software (Zotero, version 7.0.15) was used for literature management. Each included article was reviewed to assess whether it was a randomized controlled trial (RCT), observational study, in vivo/in vitro experiment, or narrative/systematic review. The majority of the articles were peer-reviewed, and several randomized clinical trials (RCTs) were identified among the included studies.

The senior authors supervised the literature review process, assessed the eligibility of selected articles, resolved disagreements among co-authors, and provided critical guidance during manuscript preparation and revision. Their role also included ensuring the methodological quality and conceptual consistency of the review.

## Figures and Tables

**Figure 1 ijms-26-07906-f001:**
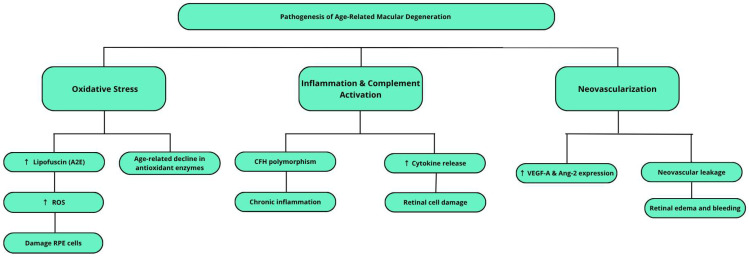
Schematic representation of the key processes contributing to the pathogenesis of AMD. The upward arrow (↑) indicates increased levels or expression.

**Figure 2 ijms-26-07906-f002:**
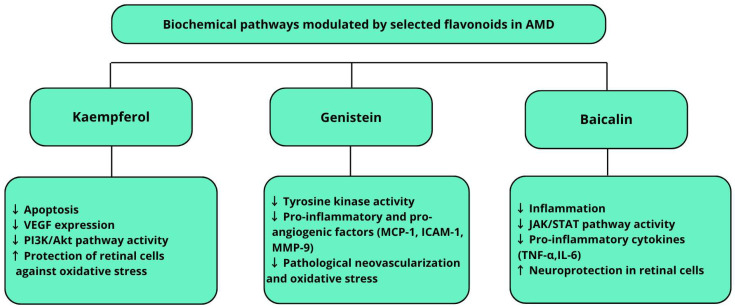
Modulatory effects of selected flavonoids on key biochemical pathways involved in age-related macular degeneration. The upward arrow (↑) indicates increased levels or expression, while the downward arrow (↓) denotes decreased levels or expression.

**Figure 3 ijms-26-07906-f003:**
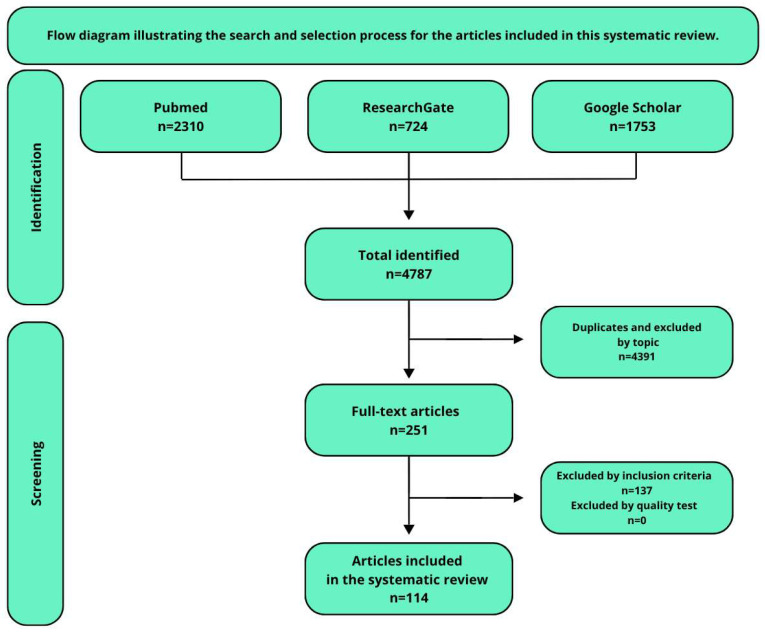
A schematic representation of the article identification, screening, and inclusion process used in this systematic review.

**Table 2 ijms-26-07906-t002:** Summary of search strategy, eligibility criteria, and study selection in the systematic review.

Criteria	Details
Databases searched	PubMed, Google Scholar, ResearchGate
Search dates	21 April 2009–4 December 2024
Search terms	“Anthocyanins in the treatment of AMD,” “AMD pathogenesis and symptoms,” “plant-derived substances used in the treatment of AMD,” “genetic background of AMD,” “Mediterranean diet in the treatment of AMD”
Language restriction	English only
Inclusion criteria	- Studies on the effects of silybin and curcumin on AMD pathogenesis - Focus on angiogenesis, oxidative stress, autophagy - Studies analyzing polyphenols and flavonoids with neuroprotective potential - Experimental, in vitro, in vivo, clinical or review studies - Molecular mechanisms involved in retinal pigment epithelium (RPE) protection
Exclusion criteria	- Non-English publications - Studies not related to AMD - Articles not addressing molecular aspects of retinal degeneration
Study types included	Randomized controlled trials (RCTs), in vitro and in vivo experiments, observational studies, narrative and systematic reviews
Peer review status	Majority of included studies were peer-reviewed
Reference management	Zotero (v7.0.15) used for duplicate removal and organization
Total articles included	114

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
