# Peer review of "The Supportive Role of Plant-Based Substances in AMD Treatment and Their Potential"

_ijms, 2025, doi:10.3390/ijms26167906_

Round 1

Reviewer 1 Report

Comments and Suggestions for Authors

This review paper gives an overview of the relevant literature about the potential role of natural plant-derived compounds in the prevention and treatment of age-related macular degeneration in the last 16 years. Generally, it is well written with clear sentences and relevant data regarding the subject. However, the overall organization of the manuscript would benefit from improvement to enhance clarity and coherence. A more structured presentation of the sections and a clearer logical flow between them would make the manuscript easier to follow and understand.

I suggest condensing the Abstract, especially the last two paragraphs that should be merged in one. The abstract looks more like an Introduction, and the proper Introduction is missing. The overall problem of the manuscript is its organization. Introduction gives only general information on AMD which should be a separate section connected with subtitles 2. and 3.

I do not think that section 4. Materials and methods are needed since it is a review article. The information can be incorporated into the Introduction, but i the authors wish to leave it like that, it should be moved at the end before or after conclusion.

Also, 7. Limitations should be moved before conclusion and shortly mentioned in it.

I do not see the point of Figure 1. It is merely a Table of Contents in the graphical form. It is also not mentioned in the text. I suggest to make a simpler graphical representation of processes connected to AMD and influenced by diverse plant substances, or similar…

There is no mention of Figure 2 in the text, as well.

Latin names should be in italic as well as “in vitro” and “in vivo”, please correct throughout the text.

Why are they termed “5.1 New plant substances with potential in AMD treatment”? Written like this it gives the idea that the substances are new. Only at the end of the manuscript, the authors give a short explanation of what is meant, which I think should be included right after this subtitle.

Why is the paragraph 5.1.1 named „Silymarin“ when all the information is given on its active component, silibinin? I suggest rewriting the beginning of the paragraph better explaining the connection between silymarin and silibinin, i.e. shortly describing the significance of silymarin.

Line 239 to which substance is “This substance…” referring to, since it is a separate paragraph from previous, an again in line 244 “…this substance...”

Line 267, please check the given references 36 and 37, those are not from the mentioned Author

Line 340, please correct the Author name

Line 491, add curcumin since it is also a polyphenol discussed in detail in this section.

The beginning of the paragraph 5.4. is repeating the general information already given more times before, therefore it is not needed.

Reviewer 2 Report

Comments and Suggestions for Authors

In this manuscript, the authors reviewed the plant derivatives in treating/preventing age related muscular degeneration (AMD). The authors summarized an extensive amount of literatures and compounds mechanism of action on AMD. The review primarily focuses on several classes of small molecules and reported studies, sheding lights on understanding of pharmacology and potential treatment for AMD. The manuscript is recommended for submission with the following minor revisions:

  1. the material and method section seems unnecessary. The ref section is good enough to demonstrate the literatures reviewed for this manuscript.
  2. It is recommended that the authors include several pictures on how each class of compound act on AMD or involved signaling pathways. A summary table is insufficient for visualization.

Reviewer 3 Report

Comments and Suggestions for Authors

The manuscript presents a comprehensive literature review exploring the potential role of plant-derived substances—particularly flavonoids, polyphenols, carotenoids, and specific compounds like silibinin and curcumin—in the prevention and treatment of age-related macular degeneration (AMD). The review is timely and addresses a clinically significant topic.

The structure is logical, the language is mostly clear, and the article demonstrates a commendable effort in integrating mechanistic insights with therapeutic implications. However, there are several aspects that require attention.

1. Scientific Content and Rigor

  • Scope and novelty: While the article is well-researched, several sections lean heavily on previously established data without sufficient critical synthesis. Aim to identify gaps in knowledge, conflicting results, or unanswered questions rather than summarizing existing work.

  • Balance: There is occasional overemphasis on in vitro and animal studies without enough acknowledgment of their limitations when translated to clinical contexts. This should be clearly stated to temper expectations.

  • Mechanistic depth: The paper does a good job discussing mechanisms like PI3K/Akt/mTOR, but many pathways are only briefly referenced without adequate diagrams or explanations that would benefit non-specialist readers.

2. Structure and Clarity

  • Redundancy: Certain mechanisms (e.g., oxidative stress, VEGF regulation) are repeated across sections. Consider consolidating these to reduce repetition.

  • Figures and Tables:

    • Figure 1 is mentioned as “work plan” but is not informative or necessary. Consider removing or replacing with a visual overview of molecular pathways.

    • Table 1 is useful but should include a column for clinical evidence (in vitro / in vivo / clinical trial).

  • Abstract: It should be rewritten to highlight specific findings and research directions. Phrases like “this review aims…” are generic; be more specific about which substances are covered and why they are significant.

3. Literature and Methodology

  • The search strategy is briefly outlined, but lacks PRISMA compliance (flow diagram is overly simplified). Clearly state inclusion/exclusion criteria and search limits in a transparent table format.

  • Provide a quality assessment of the included studies—were they randomized? Peer-reviewed? What level of evidence do they offer?

4. Language and Grammar

  • Numerous grammatical errors and awkward phrasings occur throughout. For instance:

    • “This substance, by neutralizing free radicals…” should be “Silibinin neutralizes free radicals…”

    • Avoid using passive voice excessively.

  • The manuscript would benefit significantly from professional language editing.

5. Ethical and Disclosure Issues

  • The authors are affiliated with a medical university, and many appear to be students. Please clarify the senior author’s role in overseeing the literature search and writing.

  • Conflict of interest and funding statements are missing and must be included.

Comments on the Quality of English Language

The English language throughout the manuscript is generally understandable, but it requires substantial revision to meet the standards of an international peer-reviewed journal. The issues are consistent and occur across all sections, including the abstract, introduction, body, and conclusion.

  • The manuscript must undergo professional language editing by a native English-speaking academic editor or an English language editing service (such as MDPI’s own service or Elsevier Language Services).

  • A strong emphasis should be placed on:

    • Improving sentence flow and structure

    • Using active voice where appropriate

    • Clarifying technical language for better scientific precision

Round 2

Reviewer 1 Report

Comments and Suggestions for Authors

The authors have addressed all the issues from the revision process. The organization of the manuscript has been substantially improved. Also, new schematic representations contributed to the quality and clarity of the manuscript. Therefore I suggest this manuscript to be accepted for publication.

Reviewer 3 Report

Comments and Suggestions for Authors

The changes made are substantial and appreciated. That said, before final acceptance can be considered, we encourage you to perform the following:

- Review and refine the bibliographic references to ensure all sources are recent, relevant, and correctly formatted.

- Carefully check that the manuscript aligns with the journal’s writing and formatting guidelines, including proper use of headings, terminology, citations, and style conventions.

- Ensure consistency in language and clarity throughout the text.